Methods

# A protocol for single nucleus RNA-seq from frozen skeletal muscle

Tyler GB Soule[1] , Carly S Pontifex[1], Nicole Rosin[1,2] , Matthew M Joel[1], Sukyoung Lee[1], Minh Dang Nguyen[1,3] ,
Sameer Chhibber[3], Gerald Pfeffer[1,3,4]

**Single-cell technologies are a method of choice to obtain vast amounts of cell-specific transcriptional information under physiological and diseased states. Myogenic cells are resistant to single-cell RNA sequencing because of their large, multinucleated nature. Here, we report a novel, reliable, and cost-effective method to analyze frozen human skeletal muscle by single-nucleus RNA sequencing. This method yields all expected cell types for human skeletal muscle and works on tissue frozen for long periods of time and with significant pathological changes. Our method is ideal for studying banked samples with the intention of studying human muscle disease.**

February 2023 | Accepted 27 February 2023 | Published online 13 March 2023

## Introduction

Skeletal muscle is a complex tissue composed of around 40% of an adult human's mass (Frontera & Ochala, 2015). It contains many cell types, including satellite cells, myotubes, myogenic precursors, fibroadipogenic progenitors, fibroblasts, ECs, pericytes, adipocytes, immune cells, and smooth muscle (De Micheli et al, 2020; Rubenstein et al, 2020; Orchard et al, 2021; Scripture-Adams et al, 2022). Single-cell sequencing technologies are becoming an important tool to interrogate the individual transcriptomes of cell types in otherwise complex heterogenous tissues. Increasingly, this technique is proving to be well suited for simultaneously examining many cell types, uncover rare disease states, and identify subpopulations with distinct functions (Birnbaum, 2018). Few studies have examined human skeletal muscle at the single-cell level (Barruet et al, 2020; De Micheli et al, 2020; Rubenstein et al, 2020; Orchard et al, 2021; Scripture-Adams et al, 2022), however, its utility is clear for basic and medical science studies of muscle in health and disease.

Many challenges exist to developing a method to profile all cell types in human skeletal muscle. Because muscle cells are one of the few multinucleated cells in the human body, there are major

limitations to using single-cell RNA sequencing (scRNAseq) on muscle. The most commonly used scRNAseq method relies on individual cells being encapsulated in a droplet during processing. Because of microfluidic size constraints of 10X and FACS machines, only cells smaller than 70 $\mu m$ can be sorted. Multinucleated muscle cells are exceedingly large and it is not possible to FACS sort them or pass them through filters or microfluidics (Zeng et al, 2016). For this reason, single-cell data typically include very low proportions of myofibers (De Micheli et al, 2020). Conversely, isolating nuclei from muscle yields very high proportions of myonuclei (Dos Santos et al, 2020; Petrany et al, 2020; Orchard et al, 2021). Single nuclear RNA sequencing (snRNAseq) is the best option if the experimental objective is to obtain sequencing data from all cell types in a muscle biopsy.

snRNAseq has been demonstrated by multiple groups to yield comparable data to scRNAseq (Zeng et al, 2016; Slyper et al, 2020). snRNAseq recovers the same cell types present in a tissue, but in different proportions (Slyper et al, 2020). Notably, there are no significant differences between gene expression profiles of nuclei and whole cells (Barthelson et al, 2007; Grindberg et al, 2013), with the exception of a few specific processes like cell cycle, mitosis, and transcription (Barthelson et al, 2007). Also, nuclei are enriched for non-coding RNAs (Zeng et al, 2016) and contain a higher proportion of introns (Grindberg et al, 2013). snRNAseq can target up to 40,000 nuclei, enabling the analysis of many nuclei at once (Orchard et al, 2021).

Maintaining the integrity of the nuclei and the RNA contained within is critical if an experiment is to yield useful data. For this reason, using freshly isolated tissue is the preferred approach for scRNAseq and snRNAseq. Obtaining fresh human muscle samples is possible but not always practical. Muscle biopsy is an invasive procedure that is usually performed for clinical and diagnostic purposes. Obtaining a fresh sample requires advance consent and extensive coordination with health providers for a highly time-sensitive protocol. Using frozen tissue negates the requirement to process biopsies as soon as they are obtained. It removes time-sensitivity for obtaining consent and sample handling, which can

[1]Hotchkiss Brain Institute, University of Calgary, Calgary, Canada   [2]Faculty of Veterinary Medicine, University of Calgary, Calgary, Canada   [3]Department of Clinical Neurosciences, Cumming School of Medicine, University of Calgary, Calgary, Canada   [4]Department of Medical Genetics, Cumming School of Medicine, University of Calgary, Calgary, Canada

Correspondence: gerald.pfeffer@ucalgary.ca

both be managed after the sample has been collected and frozen. This makes banked tissues available for snRNAseq. The use of frozen tissues also optimises resource utilisation, because the decision to pursue snRNAseq may be made after additional information is available from tissue histopathologic or other molecular studies.

Few articles have applied snRNAseq to skeletal muscle (Zeng et al, 2016; Dos Santos et al, 2020; Jiang et al, 2020; Kim et al, 2020; Petrany et al, 2020; Orchard et al, 2021; Eraslan et al, 2022). Of these, three groups isolated nuclei directly from mouse muscle (Dos Santos et al, 2020; Kim et al, 2020; Petrany et al, 2020), two used cell cultures (Zeng et al, 2016; Jiang et al, 2020), and four used human skeletal muscle (Orchard et al, 2021; Eraslan et al, 2022; Perez et al, 2022; Scripture-Adams et al, 2022), although Orchard and colleagues focussed on analyzing accessible chromatin regions by means of ATAC-seq. To isolate nuclei directly from muscle, these reports used fiber dissection and dounce homogenization (Dos Santos et al, 2020; Scripture-Adams et al, 2022), or a mix of various homogenization methods and detergents (Petrany et al, 2020; Orchard et al, 2021; Eraslan et al, 2022). One recently published work used fresh muscle samples with a proprietary instrument specifically designed for sc/snRNAseq (S2 Genomics) (Perez et al, 2022).

Here, we report a simple, fast, and reliable method to isolate over 7,000 nuclei on average, an excellent yield comparable to other methods (Dos Santos et al, 2020; Kim et al, 2020; Petrany et al, 2020; Orchard et al, 2021; Scripture-Adams et al, 2022), and isolate nuclei from muscle that has been frozen for up to 15 yr. Our protocol uses mechanical disruption, filtration, and fluorescence-assisted nuclei sorting (FANS) to preserve nuclear membrane and RNA integrity. The method can be applied to different human muscle types in healthy and diseased states.

# Results

## Consistent quality control on a variety of samples

Many sequencing, mapping, and cell metrics were consistently high across all samples (Table 1). All sequencing metrics exhibited little variation between samples. Only sample M1 did not meet the 10X recommended minimum of 70% fraction of reads in nuclei. Given this, our average fraction of reads in nuclei was high (83%), suggesting low ambient RNA in the sample (Hong et al, 2022). This agrees with our microscopy observations, which demonstrated intact nuclei with little to no blebbing and minimal clumping (Fig 1). All samples meet 10X Genomics recommended minimum of 20,000 mean reads per nuclei and recommended greater than 30% reads mapped confidently to the transcriptome. We observed a large number of reads corresponding to intronic regions (54%); however, this was expected because we are sequencing nuclear RNA.

We isolated between 12,000 and 27,000 nuclei per sample with an average of 19,500. Only two samples yielded less than 16,000 nuclei, in which case, we loaded the full suspension volume onto the 10X Chromium Controller. Although we targeted 10,000 nuclei, the number of nuclei recovered after sequencing varied. Our two

controls recovered around 10,000 nuclei, with the disease samples yielding less (respectively, 1,161, 3,944, 6,997, 8,531). This is likely because of the increased fibroadipose tissues that are contained within the diseased muscle. Despite this, quality control metrics were similar between the diseased muscle and controls. Furthermore, our values were similar to previous studies (Barruet et al, 2020; Dos Santos et al, 2020; Kim et al, 2020; Petrany et al, 2020; Scripture-Adams et al, 2022), although quality control metrics were not always reported in full in these works. In summary, quality control metrics meet 10X Genomics' recommendations and are in line with values from similar studies.

## Cell populations are consistent with previous findings

In total, we recovered 43,325 nuclei from six skeletal muscle biopsies. We performed unbiased clustering on all nuclei (Fig 2). We observed read and mitochondrial DNA distribution throughout each cluster, indicating that clustering was not strongly influenced by the number of reads per nucleus or mitochondrial DNA content (Fig S1).

We demonstrated that this method recovers cell populations expected in skeletal muscle (Fig 2). Canonical markers and top differentially expressed genes (Table S1) were used to assign the identity of each cluster (Fig 3). Overall, we identified 12 cell types whose identities are consistent with previous findings (De Micheli et al, 2020; Rubenstein et al, 2020; Orchard et al, 2021; Scripture-Adams et al, 2022). For assignment of Type I and Type II fibers, we used previously reported gene sets (Rubenstein et al, 2020), which clearly separated Type I and Type II fibers (Fig 4). Genes reported by Rubenstein et al to be differentially expressed between Type I and Type II fibers were highly correlated to our values for the same genes (Fig S2). We observed a third type of muscle in every sample which expressed both Type I and Type II muscle genes. Based on the expression of genes only present in differentiating muscle like NCAM1 (Capkovic et al, 2008), COL19A1 (Sumiyoshi et al, 2001), and MYH3 (Beylkin et al, 2006), we concluded that these represent differentiating myonuclei.

Cell type proportions were very similar between controls C1 and C2, despite originating from different anatomical locations (Fig S3). Interestingly, samples m1 and M1 also closely matched the recovered cell type proportions of the controls. Cell type proportions in M2 and M3 varied from the controls and from each other. This was expected from samples with major histopathologic changes. Also, M2 and M3 still contained the same cell types, just in different proportions. On average, myonuclei and fibroadipogenic progenitors represented 82% of all nuclei. All other cell types made up a small proportion of the total cell number and displayed minor variations between samples.

Previous reports identified lymphatic ECs in human muscle (Eraslan et al, 2022; Scripture-Adams et al, 2022) and mouse muscle (Feng et al, 2019). Gene pathway analysis of the top 200 marker genes in our second EC cluster showed significant enrichment of lymphatic EC differentiation, blood vessel EC differentiation, and lymphangiogenesis (Fig S4). Our lymphatic ECs shared 71% and 78% of top marker genes with Scripture-Adams et al and Eraslan et al, respectively (Eraslan et al, 2022; Scripture-Adams et al, 2022). We concluded that this cluster represents lymphatic ECs. We also identified mast cells by canonical markers and GO analysis,

**Table 1.** Quality control metrics: quality control data represent six samples across two separate experiments. The average and SD for all metrics is represented.

| Sequencing metrics | | | | | | |
|---|---|---|---|---|---|---|
| Sample | No. of reads | Valid barcodes (%) | Valid UMIs (%) | Sequencing saturation (%) | Q30 bases in barcode (%) | Q30 bases in RNA read (%) | Q30 bases in UMI (%) |
| C1 | 229,291,982 | 96.7 | 99.9 | 72.7 | 96.4 | 93.7 | 96.2 |
| C2 | 268,498,276 | 96.2 | 100 | 36.9 | 95.2 | 92.1 | 94.9 |
| m1 | 243,061,000 | 96.6 | 99.9 | 79 | 96.3 | 93.1 | 96.2 |
| M1 | 110,311,933 | 95.2 | 99.9 | 91.8 | 96.4 | 94.5 | 96.3 |
| M2 | 252,256,865 | 97.3 | 99.9 | 88.6 | 96.5 | 94 | 96.4 |
| M3 | 226,248,973 | 96.2 | 100 | 37.7 | 95 | 91.9 | 94.7 |
| MEAN | 221,611,504.8 | 96.4 | 99.9 | 67.8 | 96.0 | 93.2 | 95.8 |
| STDV | 56,684,528.4 | 0.7 | 0.1 | 24.6 | 0.7 | 1.0 | 0.8 |
| Read mapping metrics | | | | | | |
| Sample | Mapped to genome (%) | Mapped confidently to genome (%) | Mapped confidently to intergenic regions (%) | Mapped confidently to intronic regions (%) | Mapped confidently to exonic regions (%) | Mapped confidently to transcriptome (%) | Mapped antisense to gene (%) |
| C1 | 96.8 | 92.9 | 6.2 | 54.8 | 31.9 | 69.4 | 15.9 |
| C2 | 92.5 | 87.7 | 7 | 61.3 | 19.4 | 44.2 | 35.4 |
| m1 | 96.7 | 93.3 | 6.3 | 56.3 | 30.6 | 67.8 | 17.7 |
| M1 | 92.9 | 65.1 | 6.6 | 40.1 | 18.3 | 52.8 | 4.1 |
| M2 | 96.2 | 91.4 | 7.5 | 54.8 | 29.1 | 72.6 | 9.4 |
| M3 | 91.4 | 86 | 7.2 | 58.9 | 19.9 | 44 | 33.7 |
| MEAN | 94.4 | 86.1 | 6.8 | 54.4 | 24.9 | 58.5 | 19.4 |
| STDV | 2.4 | 10.7 | 0.5 | 7.4 | 6.3 | 13.0 | 12.7 |
| Cell metrics | | | | | | |
| Sample | Recovered nuclei | Fraction reads in nuclei (%) | Mean reads per nucleus | Median genes per nucleus | Total genes detected | Median UMI counts per nucleus | |
| C1 | 9,697 | 89.4 | 23,345 | 1,409 | 26,472 | 3,232 | |
| C2 | 13,030 | 90.8 | 20,565 | 1,981 | 28,120 | 4,729 | |
| m1 | 8,531 | 91.1 | 28,279 | 1,344 | 26,198 | 3,008 | |
| M1 | 1,161 | 60.6 | 87,914 | 771 | 19,440 | 1,221 | |
| M2 | 3,944 | 79.1 | 61,406 | 1,533 | 26,062 | 2,730 | |
| M3 | 6,997 | 86.9 | 32,183 | 2,469 | 28,497 | 6,654 | |
| MEAN | 7,226.7 | 83.0 | 42,282.0 | 1,584.5 | 25,798.2 | 3,595.7 | |
| STDV | 4,221.5 | 11.8 | 26,744.4 | 581.8 | 3,279.8 | 1,871.0 | |

which showed mast cell activation, gastric acid secretion, molecular mediation of the inflammatory response, and Fc-ε receptor signalling pathways (Fig S4).

## Discussion

Here, we have demonstrated a practical and effective method of isolating nuclei from human skeletal muscle. Our method yields sufficient quantities of nuclei for snRNAseq. In total, 43,325 nuclei were recovered. With this method, we identified 12 distinct cell types which are consistent with previous reports in human skeletal muscle (De Micheli et al, 2020; Rubenstein et al, 2020; Orchard et al, 2021). Notably, this included a large proportion of type I and type II myonuclei. To our knowledge, there have been only a few reports of successfully isolating human myonuclei in significant quantities (Orchard et al, 2021; Eraslan et al, 2022; Scripture-Adams et al, 2022).

Importantly, we were able to obtain consistent quality control metrics from a variety of samples. This included muscle representing both sexes, collected from the commonly biopsied deltoid and vastus lateralis muscles, and from patients from 33 up to 64 yr of age. In addition, the biopsy from patient M2 was performed in 2006 and still yielded 4,000 nuclei (Table 1), suggesting that this

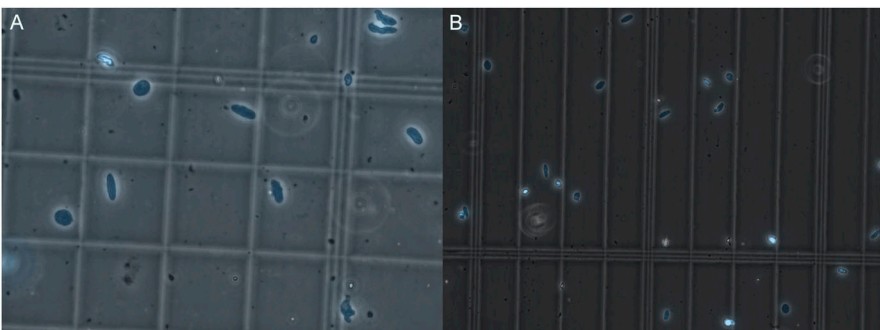

**Figure 1. Nuclear quality visualized by microscopy.**
**(A, B)** Representative images of nuclei quality before (A) and after (B) filtration and FANS sorting of nuclei. Nuclei are marked in blue by DAPI staining. Three images were taken for every sample, and each was observed at 40X magnification as recommended by 10X to look for nuclei integrity and the absence of clumping. After sorting, the debris is reduced, nuclei are intact, and display minimal to no clumping.

method is robust to extended periods of freezing as is typical of banked samples. We included four distinct muscle disease samples to test the applicability of this method to studying muscle disease. Our results suggest that this method is a good choice for studying banked samples and rare muscle diseases, where maximizing information gained from the sample is key. We recommend that muscle samples used for this protocol should be flash-frozen after devitalisation and stored continuously in a sealed container at −80°C thereafter. Storage at temperatures above −80°C or in an

unsealed container may affect sample integrity and should be avoided. Freeze–thaw cycles are also expected to negatively impact sample integrity (Kellman et al, 2021). However, we did not formally test these recommendations in this work, and it is possible that samples with suboptimal storage could yield useable results in the course of future study (de Oliveira et al, 2012).

Each tissue requires specific protocol adaptations to maximize the yield and quality of nuclei (Slyper et al, 2020; Eraslan et al, 2022). Common elements that vary between protocols include buffers,

**Figure 2. Cell types detected by snRNAseq in human skeletal muscle.**
UMAP showing 43,325 nuclei separated by unbiased clustering and labeled by cell type.

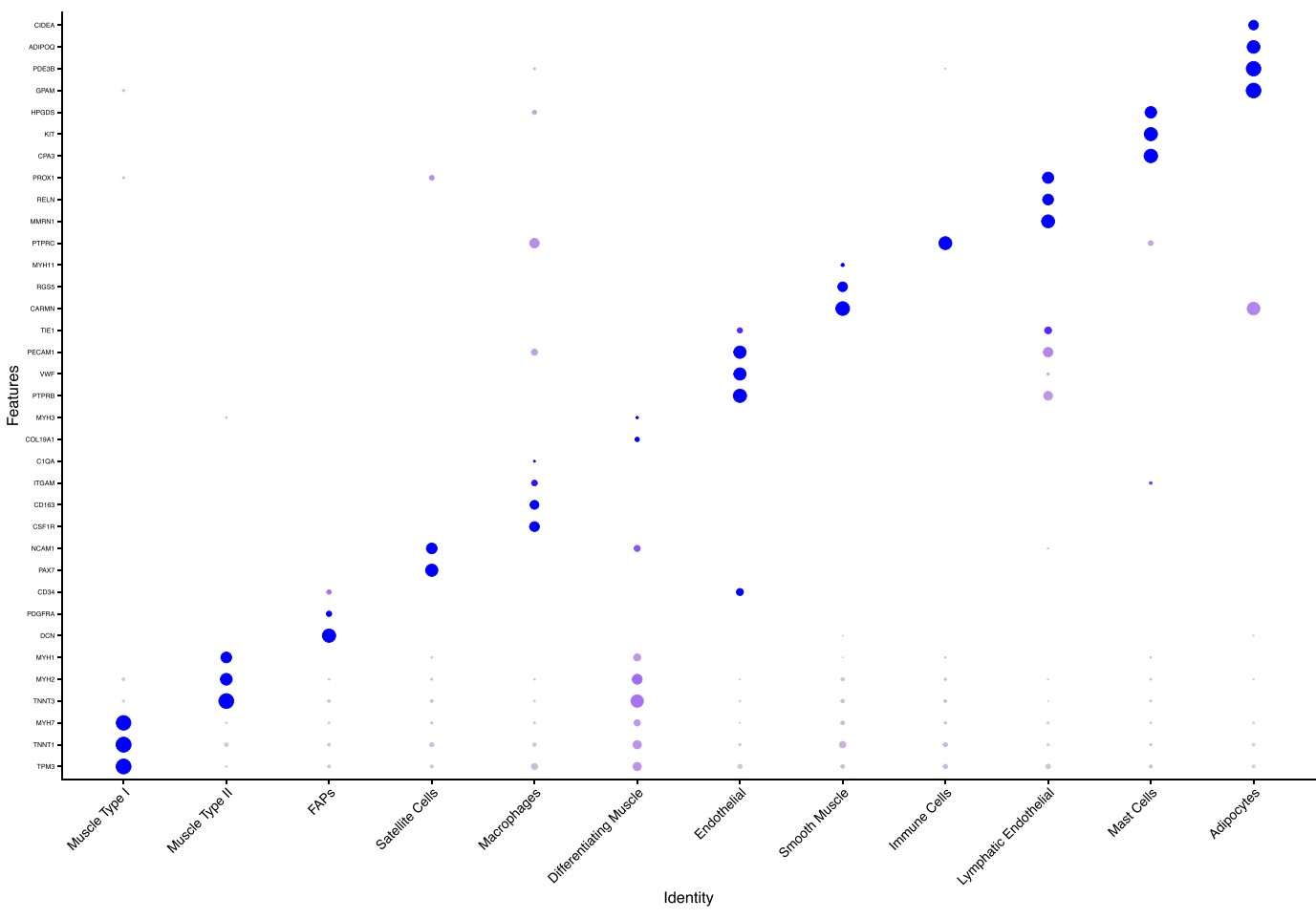

**Figure 3. Marker genes used to assign cell type.**
Dot plot showing the expression of genes used to assign each cluster. Dot size indicates the percentage of cells expressing the gene and dot colour indicates the average gene expression.

instruments of dissociation, and methods of purification. This is emphasized by the distinct protocols developed to isolate nuclei from the liver (Cavalli et al, 2020), kidney (Leiz et al, 2021), and tumours (Narayanan et al, 2020; Slyper et al, 2020). We have developed a protocol that works for the dense, fibrous nature of skeletal muscle. Table 2 illustrates several advantages to our method (with the caveat that methods and resulting from single-cell and single-nucleus isolation will not always be directly comparable). Specifically, ceramic beads avoid the difficulties associated with Dounce homogenization (Fig S5). As recommended by 10X Genomics, FANS was used to purify the nuclei because it minimally impacts RNA expression and integrity compared with enzymatic dissociation. The use of enzymes like collagenase and dispase are widely used to dissociate muscle tissue for single-cell analysis, but have been reported to affect transcription in single cells (Van Den Brink et al, 2017; Orchard et al, 2021; Eraslan et al, 2022). Overall, we have developed a method that minimizes the time between sample homogenization and gel beads in emulsion (GEM) generation, without sacrificing RNA integrity.

Notably, we did not find a strong correlation between the mass of homogenized muscle and quantity of recovered nuclei (Fig S6),

suggesting either sample-specific effects or an effect of the ratio of lysis buffer to muscle mass. Regardless, we observed that masses of around 55–70 mg were ideal to avoid excess fibrous tissue that interfered with pipetting and filtering (Video 1). In addition to filtering, FANS was an effective method to get rid of excess debris that could interfere with downstream processing and microfluidics.

Two samples recovered significantly less nuclei than controls. We speculate that sample integrity could be compromised because of dead and dying cells already present at the time of tissue collection. This was illustrated by sample M2, which demonstrated a slightly lower percentage of reads in the nuclei (79.1%) and less overall recovered nuclei (3,944) (Table 1). Sample M2 was biopsied in 2006, so freezing time could also play a role. Although quality control was similar to our other samples, further study is required to determine if age of muscle biopsy affects nuclei yield or if insufficient nuclei were released during homogenization. In addition, sample M1 had 60% of reads in nuclei and recovered 1,255 nuclei. High reads per nucleus (87,914) and high sequencing saturation (91.8%) suggested a low complexity library. Notably, other QC metrics for both samples such as mean reads per nucleus, median genes per nucleus, and total genes detected were on par with

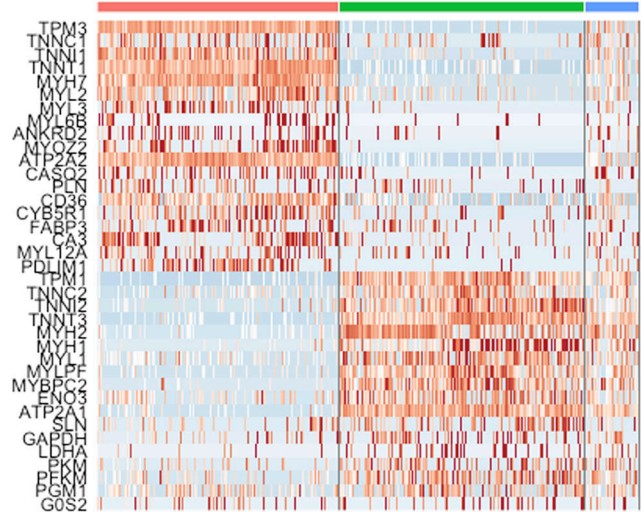
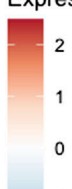

**Figure 4. Differentiating muscle fibers by gene expression.**
Heat map using genes identified by Rubenstein and colleagues to distinguish type I and type II fibers (Rubenstein et al, 2020). Mature type I and II myonuclei clearly express type I or II genes, whereas differentiating myonuclei express many genes from both fiber types.

controls. None of the samples underwent freeze–thaw cycles before nuclei isolation, so this variable could not impact the quality of recovered nuclei. In addition, it is unlikely that the isolation protocol induced a comparatively higher level of nuclei damage in these samples as they were processed side by side with other samples that recovered more nuclei. Overall, these results suggest that diseased samples with compromised tissue might reduce the total number of isolated nuclei, while still giving good quality data for the successfully recovered nuclei.

Although skeletal muscle contains type I, type IIa, type IId/x, and type IIb fibers (Hawley et al, 2014), we designated them as either type I or type II. This is because we are aware of only one study that has characterized transcriptional differences between fiber types in human muscle (Rubenstein et al, 2020). Therefore, although we can only accurately distinguish between these two fiber types, it is obvious which nuclei represent type I and II muscles (Fig 4). In addition, there was a high degree of correlation when comparing

differentially expressed genes between type I and type II fibers identified by Rubenstein et al (Fig S2). This gives confidence in accurate myonuclei assignments and demonstrates consistency between methods. Overall, a more detailed characterization of transcriptional variation between human muscle subtypes could lead to a more accurate classification of snRNAseq-derived data.

Proportions of type I, type II, and differentiating myofibers were relatively consistent between all samples except for M2 (Fig S3). Some variability between myofiber types was expected because of age (McCormick & Vasilaki, 2018), variable exercise regimes of participants (Hawley et al, 2014), and muscle-specific differences. In addition, selective atrophy of either fast or slow type muscle fibers has been reported for various muscle diseases (Wang & Pessin, 2013) andsarcopenia (Evans & Lexell, 1995). Other cell types also vary in number between people. For example, many factors influence the number of satellite cells in muscle tissue including age (Chen et al, 2020; Yablonka-Reuveni, 2011), specific muscle location

**Table 2. Method comparison: justification of technical and practical advantages to our methodology.**

| Previous methods | Our method | References |
|---|---|---|
| **scRNAseq** | **snRNAseq** | Zeng et al (2016); De Micheli et al (2020); Dos Santos et al (2020); Slyper et al (2020) |
| Requires fresh tissue | Allows analysis of frozen banked samples | |
| Large, multinucleated myofibers cannot pass through microfluidics | Nuclei from all cell types can pass through microfluidics | |
| Very few myofibers represented | High proportion of myonuclei present | |
| **Enzymatic dissociation** | **FANS** | Chongtham et al (2021); Orchard et al (2021); Van Den Brink et al (2017) |
| Can influence the transcriptional behaviour of muscle satellite cells | Increase the number of nuclei obtained | |
| Typically requires an hour of enzymatic action | Minimal, if any impact on library quality | |
| **Dounce homogenization** | **Lysing using a bead tube** | Martelotto and Martelotto (2020) |
| Human muscle is too fibrous, yields too much debris, and results in poor filtering capacity and nuclei yields (Fig S5) | Ceramic beads and 5-s pulses of high rpm shaking break down fibrous material | |
| Relatively slow process Inter-operative variability | Relatively fast process consistent timing and shaking speed | |

(Yablonka-Reuveni, 2011), fibre diameter (Maier & Bornemann, 1999), muscle type (I versus IIa, IIx, and IIb) (Yin et al, 2013), disease (Chang et al, 2016), inflammation, exercise, and injury (Chen et al, 2020; Yablonka-Reuveni, 2011). In addition, the number of satellite cell nuclei recovered in the isolation procedure may depend on the extent of tissue mincing. Finally, variation between cell type proportions has been reported in single-cell analysis of human muscles biopsied from different anatomical locations (De Micheli et al, 2020), and in mice at different ages (Petrany et al, 2020). Overall, care should be taken when interpreting cell type proportions present in a human muscle biopsy, as they are influenced by many factors.

Many cell types have been hypothesized to be involved in muscle disease, such as satellite cells in Duchenne muscular dystrophy (Chang et al, 2016), fiboradipogenic progenitors in muscular dystrophies, other myopathies, and aging (Molina et al, 2021; Theret et al, 2021), and various immune populations as both integral parts of injury response and pathology when dysregulated (Chen & Shan, 2019; Farini et al, 2021; Venalis & Lundberg, 2014). However, we believe that examining myogenic cells is also crucial to gain a full understanding of the muscle environment. Importantly, we obtained many myonuclei using this technique, representing an average of 67% of our total nuclei. Interestingly, our ECs formed two distinct clusters, one of which we concluded was lymphatic ECs (Fig 2). Our lymphatic ECs shared a high percentage of top marker genes with Scripture-Adams et al and Eraslan et al, respectively. Despite Feng and colleagues working with data from mouse ECs, 28% of their marker genes were also present in our lymphatic EC top marker genes, including *MMRN1* and *PROX1*. Interestingly, like Feng and colleagues' findings, *MMRN1* was also a specific marker for lymphatic ECs in our human muscle samples (Fig S4). We also identified mast cells. Mast cells have been reported in human skeletal muscle before (Rubenstein et al, 2020) as part of a cluster of myeloid cells, which are transcriptionally similar. We conclude that our cluster is predominantly mast cells. Our mast cell markers *CPA3*, *KIT*, and *HPGDS* were expressed in 90%, 88%, and 78% of cells in this cluster, and the GO analysis points towards several mast cell-specific processes (Fig S4). In our samples this cell type was present in two of four pathologically affected muscles, but in higher proportions than controls (Fig S3). There is some evidence that mast cells are enriched in inflammatory myopathies (Yokota et al, 2014), Duchenne muscular dystrophy (Gorospe et al, 1994) and cachectic muscle (Widner et al, 2021). These results emphasize this method's ability to detect rare cell types and enable further investigation into their behaviour in pathological conditions.

Our method has the potential to elucidate transcriptional responses of muscle cell types in disease, exercise, and homeostatic conditions. We identified all expected cell types, and mast cells and lymphatic ECs. Although this emphasizes the robust nature of our approach, we only analyzed biopsies from two anatomical locations. Additional rare cell types could be uncovered in more diverse muscle biopsies. Future studies of banked samples with a variety of sexes, ages, and diseases would be valuable. In the course of future study, it would also be informative to perform a direct comparison of different methods to more clearly outline the advantages of each protocol without inter-operator variability.

# Materials and Methods

### Human skeletal muscle sample collection

Research ethics board approval was obtained for this study from the University of Calgary Conjoint Health Research Ethics Board (REB16-2196). Additional muscle samples for research were collected during clinical muscle biopsy procedures, with advance written informed consent. We also used banked muscle tissues from prior procedures, again with written informed consent (REB15-2763). Samples were stored at −80°C after devitalisation. We included two samples from participants with normal histopathology and 4 samples from participants with myopathic abnormalities (Joyce et al, 2012) (m1 having minor abnormalities and M1, M2, and M3 having more severe abnormalities, Table 3).

### Materials

The requisite materials and reagents used for this protocol are listed (along with suppliers and product numbers) in Table 4.

### Preparation

- Set a swinging bucket rotor to 4°C (fixed bucket rotors may result in loss of nuclei to the sides of the tube rather than collecting the nuclei as a pellet at the base of the tube)
- Prepare nuclei lysis buffer (see the Materials and Methods section) at a minimum of 750 µl per sample
- Prepare nuclei wash and resuspension buffer (see the Materials and Methods section) at a minimum of 550 µl per sample
- Prepare nuclei wash and resuspension buffer (see step 14) for sorting at a minimum of 100 µl per sample.

### Tissue homogenization

Note: Maintain samples on ice wherever possible. All steps should be performed using a 1-ml pipette with a cut tip to create a wider bore unless otherwise indicated. Wide bore tips may be purchased (see the Materials and Methods section) or generated by cutting the tip of a standard 1,000-µl pipette tip to make a bore ~1.3 mm in diameter.

(1) On a petri dish in a 4°C room, cut ~60 mg of muscle tissue and mince with a sterile scalpel until muscle is reduced to a slurry.
(2) Transfer the muscle to a 2 ml bead tube and add 500 µl Nuclei Lysis buffer (see the Materials and Methods section).
(3) Secure the bead tube in a mechanical homogenizer and shake at 3,000 rpm for 2 × 5 s with a 5 s break in between cycles on ice.
(4) Pipet off the supernatant into a Lo-Bind 1.5 ml Eppendorf. Ensure to minimize residual tissue fragments transferred with the supernatant as this will interfere with filtration.
(5) Add 250 µl Nuclei Lysis buffer to the tube containing residual tissue and lyse again for 10 s at 3,000 rpm. Repeat step 4.
(6) The degree of lysis will depend on tissue composition of the sample, as fibrotic or adipose tissue content can be variable. There may also be variation based on which muscle and what part of the muscle is biopsied. Additional lysis steps may be performed if large

**Table 3. Summary of human samples: chosen samples represent both sexes, a range of ages, and various disease states of muscle.**

| Sample | Disease state | Sex | Age at biopsy | Year of biopsy | Muscle |
|--------|---------------|-----|---------------|----------------|--------|
| C1 | Healthy control | M | 48 | 2019 | Deltoid |
| C2 | Healthy control | F | 33 | 2018 | Vastus lateralis |
| m1 | Mild myopathic changes | M | 64 | 2019 | Deltoid |
| M1 | Major myopathic changes | F | 52 | 2018 | Vastus lateralis |
| M2 | Major myopathic changes | M | 59 | 2006 | Deltoid |
| M3 | Major myopathic changes | M | 36 | 2018 | Deltoid |

tissue fragments remain, however, the processing time before GEM generation should be minimized to avoid RNA degradation. Perform preliminary assessments to determine the number of lysis steps needed to isolate sufficient nuclei.

### Nuclear isolation

(7) Transfer the remaining clear supernatant to the Lo-Bind tube.
(8) Centrifuge in a swinging bucket rotor at 500$g$, 5 min, 4°C. You should see a large pellet containing debris and nuclei.
(9) To avoid disturbing the pellet, carefully pipet the supernatant using a P200. Discard the supernatant.
(10) Add 200 $\mu$l nuclei wash and resuspension buffer to the pellet without resuspending and wait 5 min to allow buffer interchange.
(11) Using a cut 1,000 $\mu$l tip, add 300 $\mu$l nuclei wash and resuspension buffer and resuspend with five slow triturations.
(12) Filter the suspension through 40 $\mu$M FlowMi Cell Strainers into a labelled 5 ml round-bottom Falcon tube.
(13) Load 10 $\mu$l of suspension onto a hemocytometer, observe for nuclear quality and approximate number (Fig 1). When counting, we average the four outside corner squares of the hemocytometer, including the nuclei touching the left and top of each square. However, we recognize that other labs may have their own process for this.

### FANS sorting

(14) Sort the nuclei into a tube with 100 $\mu$l of Nuc W+R and 4 U/$\mu$l RNase inhibitor. This step is necessary because FANS sorting will dilute the nuclei wash and resuspension buffer, resulting in a low RNase inhibitor concentration. The Nuc W+R buffer and high RNase inhibitor concentration will prevent the nuclei from sitting on ice for extended periods of time without protection from RNA degradation. ***4 U/$\mu$l RNase inhibitor concentration assumes that the final nuclei suspension volume will be 2 ml after FANS sorting. Perform preliminary assessments to determine a suitable RNase inhibitor concentration for sorting.
(15) We gated the nuclei by SSC-A/FSC-A for granularity and size. Singlets were then gated based on size P2 (FSC-H/FSC-A) and granularity P3 (SSC-H/SSC-A). Finally, the P4 population was sorted for Pacific blue positive nuclei (Fig S7).
(16) Transfer the sorted nuclei to a Lo-Bind tube and centrifuge in a swinging bucket rotor at 500$g$, 5 min, and 4°C.
(17) Carefully remove the supernatant with a P200, leaving a small amount of buffer near the bottom to avoid losing nuclei.

(18) Resuspend slowly in 50–100 $\mu$l nuclei wash and resuspension buffer.
(19) Load 10 $\mu$l of undiluted suspension onto a hemocytometer, observe for nuclear quality and count to determine nuclei concentration.

NOTE: At this stage, there should be minimal debris in the solution with the nuclei. Ideally, the nuclei should have an intact membrane and no blebbing. Ideal concentration ranges from 700–1,200 nuclei/$\mu$l. If the nuclei are too concentrated, dilute in more Nuc W+R buffer. If the nuclei are not concentrated enough, centrifuge at 500$g$, 5 min, 4°C and resuspend in a smaller volume. This will result in some nuclei loss. Alternatively, load a larger volume of nuclei suspension onto the 10X chromium device. The nuclei are now ready to proceed with 10X Genomics library preparation.

### Library preparation and sequencing

We adjusted the volume of nuclei suspension added to the 10X Genomics Chromium Controller to target 10,000 recovered nuclei. This number accounted for the expected ~65% loss of nuclei during library preparation. Library preparation is detailed by 10X Genomics in Chromium Single Cell 3' Reagent Kits User Guide (v3.1 Chemistry; Dual Index). Briefly, nuclei were suspended in GEMs, barcoded, subjected to reverse transcription, and excess reagents are cleaned up. cDNA is amplified (12 cycles), quantified by TapeStation, and assembled into a Chromium Single Cell 3' Gene Expression Dual Index Library (14 amplification cycles), and SPRIselect beads are used to size select the final cDNA libraries. Libraries were sequenced on the NovaSeq using SP 100 reagents to generate 800 million total read pairs. Read lengths were indicated by 10X Genomics as follows: read 1: 28, i7 index: 10 bp, i5: index 10 bp, and read 2: 90 bp.

### Library analysis and quality control

Samples were demultiplexed using Cell Ranger v5.0.0. After demultiplexing, the samples were aligned to a reference genome using Cell Ranger count, STAR alignment, and the reference transcriptome GRCh38-3.0.0. Because nuclei were the source of RNA, there is a high prevalence of pre-mRNA in the samples. Therefore, the "include-introns" option was used during Cell Ranger count.

Data processing, quality control, and analysis were performed in R using Seurat. Nuclei were filtered based on the number of unique genes and mean reads per nucleus. Genes expressed in fewer than 3

**Table 4. Summary of materials and reagents with suppliers and product numbers.**

| Reagent or resource | Source | Identifier |
|---|---|---|
| Consumables | | |
| Polystyrene Petri dishes | VWR | 25384-088 |
| Stainless steel disposable scalpels | Integra | 4-420 |
| 2 ml 1.44 mm ceramic bead tubes | Bertin | P000912-LYSK0-A |
| 40 $\mu$M FlowMi Cell Strainers | SP Bel-Art | 136800040 |
| RNase OUT Ribonuclease Inhibitor | Invitrogen | 10777-019 |
| 5 ml round-bottom tube | Falcon | REF 352058 or REF 352063 |
| DAPI (10 $\mu$g/ml) | Sigma Aldrich | D9542 |
| RNase away | | |
| 1,000 $\mu$l wide-bore pipet tips | Sigma-Aldrich | AXYT1005WBC |
| 1.5 ml LoBind tubes | VWR | CA80077-232 |
| Single-cell commercial assay | | |
| Chromium Next GEM Chip G Single Cell Kit, 16 rxns | 10X Genomics | 1000127 |
| Dual Index Kit TT Set A,96rxn | 10X Genomics | 1000215 |
| Chromium Next GEM Single Cell 3′Kit v3.1, 4 rxns | 10X Genomics | 1000269 |
| Library Construction Kit, 4 rxns-1000196 | 10X Genomics | 1000196 |
| Tube, Dynabeads MyOne SILANE | 10X Genomics | 2000048 |
| 10X Chromium Controller | 10X Genomics | 1000202 |
| Equipment | | |
| Neubauer Hemocytometer | VWR | 15170-208 |
| Minilys Homogenizer | Bertin | |
| FACSaria Fusion | BD | |
| ST 40R Centrifuge | Sorvall | |
| TX-750 Rotor | Thermo Fisher Scientific | |
| Buffers | | |
| Nuclei EZ lysis buffer with RNase inhibitor | Invitrogen | 10777019 |
| Nuclei lysis buffer 0.1X Nuclei EZ lysis buffer in 1X PBS with 0.2 U/$\mu$l RNase inhibitor | | |
| Nuclei wash and resuspension buffer | | |
| 1X PBS, 1.5% BSA, 0.2 U/$\mu$l RNase inhibitor, 10 $\mu$g/ml | | |
| DAPI | | |
| Software and algorithms | | |
| CellRanger | 10X Genomics | Version 6.1.2 |
| STAR | Dobin et al (2013) | Version 2.7.2a |
| R | R Core | Version 4.1.2 |
| Rstudio | R Core | Version 1.4.1106 |
| Seurat | Hao et al (2021) | Version 4.1.0 |

nuclei were not included in the downstream analyses. Nuclei with fewer than 50 genes or higher than 2% mitochondrial DNA content were also excluded, as they are likely from dying cells or the membrane was ruptured. Outlier nuclei were excluded based on examination of n_FeatureRNA versus n_CountRNA plots for combined samples. These measures serve to exclude doublets, artifacts, and poor-quality nuclei.

Samples were integrated after the Seurat vignette (Stuart et al, 2019). Gene expression was log normalized and scaled. Dimensionality reduction was achieved by using the principal component that represented over 90% of variation and less than 0.1% variation from the previous principal component. Nuclei populations and subpopulations were identified using canonical marker genes identified in the literature.

## Data Availability

Because of privacy restrictions, raw sequencing reads generated during this study are not publicly available.

## Supplementary Information

## Acknowledgements

TGB Soule is the recipient of a Hotchkiss Brain Institute Graduate Recruitment Scholarship. CS Pontifex is the recipient of an Eyes High Scholarship from the University of Calgary. S Lee is a recipient of Canada Graduate Scholarship from the Canadian Institutes of Health Research. Infrastructure for this work was supported by a John R Evans infrastructure grant from the Canada Foundation for Innovation to G Pfeffer. MD Nguyen is supported by a CIHR project grant. Nuclei were sorted by the University of Calgary Flow Cytometry Core Facility. Libraries from the nuclei were created by Dr. Rosin and the laboratory of Dr. Jeff Biernaskie. Libraries were sequenced at the University of Calgary Centre for Health Genomics and Informatics, which is supported by the Cumming School of Medicine. Cellranger was supported by resources from the Research Computing Services group at the University of Calgary.

### Author Contributions

TGB Soule: conceptualization, data curation, formal analysis, investigation, methodology, and writing—original draft.
CS Pontifex: investigation, methodology, and writing—review and editing.
N Rosin: investigation, methodology, and writing—review and editing.
MM Joel: formal analysis, methodology, and writing—review and editing.
S Lee: resources and investigation.
MD Nguyen: resources and writing—review and editing.
S Chhibber: conceptualization and resources.
G Pfeffer: conceptualization, resources, supervision, funding acquisition, investigation, and writing—review and editing.

### Conflict of Interest Statement

The authors declare that they have no conflict of interest.

### Ethics statement

Work described in this study was approved by the University of Calgary Conjoint Health Research Ethics Board, REB16-2196.

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
