## [Reviewer comments · Life Science Alliance]

Life Science Alliance

A Protocol for Single Nucleus RNAseq from Frozen Skeletal Muscle

Tyler Soule, Carly Pontifex, Nicole Rosin, Matthew Joel, Sukyoung Lee, Minh Nguyen, Sameer Chhibber, and Gerald Pfeffer
DOI: <https://doi.org/10.26508/lsa.202201806>

Corresponding author(s): Gerald Pfeffer, University of Calgary

Review Timeline:

Submission Date:	2022-11-07
Editorial Decision:	2022-12-13
Revision Received:	2023-01-04
Editorial Decision:	2023-01-20
Revision Received:	2023-02-07
Editorial Decision:	2023-02-08
Revision Received:	2023-02-24
Accepted:	2023-02-27

Scientific Editor: Novella Guidi

Transaction Report:

December 13, 2022

Re: Life Science Alliance manuscript #LSA-2022-01806-T

Dr. Gerald Pfeffer
University of Calgary
Hotchkiss Brain Institute, Department of Clinical Neurosciences
HMRB 155, 3330 Hospital Dr NW
Calgary, Alberta T2N 4N1
Canada

Dear Dr. Pfeffer,

Thank you for submitting your manuscript entitled "A Protocol for Single Nucleus RNAseq from Frozen Skeletal Muscle" to Life Science Alliance. The manuscript was assessed by expert reviewers, whose comments are appended to this letter. We invite you to submit a revised manuscript addressing the Reviewer comments.

Thank you for this interesting contribution to Life Science Alliance. We are looking forward to receiving your revised manuscript.

Sincerely,

B. MANUSCRIPT ORGANIZATION AND FORMATTING:

Reviewer #1 (Comments to the Authors (Required)):

In the submitted manuscript, Soule and colleagues describe and contextualize a protocol for fast and reliable isolation of nuclei from whole froze human skeletal muscle tissue and performed snRNASeq for sample comparison and quality control.

The protocol steps are well described although some improvements could be done to ameliorate clearness and usage from non-expert users.

INTRODUCTION

Skeletal muscle is a complex tissue composed of around 40% of an adult human's mass: please insert proper citations

It contains many cell types, including satellite cells, myotubes, myogenic precursors, fibroadipogenic progenitors (FAPs), fibroblasts, endothelial cells (ECs), pericytes, adipocytes, immune cells, and smooth muscle.: please insert proper citations

MATERIALS AND METHODS

TABLE 1: please indicate also Date of collection/Freezing of the samples

Lo-Bind 1.5 mL Eppendorf.: this material was not present in the material table

NUCLEI ISOLATION

STEP 10: it is not clear if 200 microliters of Nuclei Wash and Resuspension buffer should be added to the pellet or to the supernatant from step 9. Moreover, since in STEP 8 it has been written that the pellet should contain debris, it is not clear if it would also contain nuclei. Please be more accurate.

STEP 13: which kind of hemocytometer is more suitable for this kind of biological material? Burkner, Thoma, Neubauer? Please also specify some tips to have an accurate counting. How many squares? How to perform mean between squares? How to calculate concentration and dilution?

FANS SORTING

STEP 14: Cells will be sitting on ice for extended periods of time without proper protection from RNA degradation - this sentence seems out of context. Which cell the sentence is referring to? At this step of the protocol there should be only nuclei in the suspension. Please rephrase the sentence

STEP 14: ***RNase Inhibitor concentration assumes 2 mL final volume of sorted nuclei. Perform preliminary assessments to determine a suitable RNase Inhibitor concentration for sorting. - also this sentence is difficult to understand. please rephrase being more accurate in describing this step

STEP 15: please add a figure about gating strategy

LIBRARY ANALYSIS AND QUALITY CONTROL

In this paragraph you often refer to "cells" rather than to "nuclei". If this protocol is for snRNASeq, it would be better to refer to nuclei. Please also check the entire manuscript.

CONSISTENT QUALITY CONTROL ON A VARIETY OF SAMPLES

The average fraction of reads in cells was high (83%), suggesting low ambient RNA in the sample.: please cite and/or refer to other available protocols describing quality parameters in single cells/nuclei sequencing

DISCUSSION

The manuscript describes a protocol to isolate and characterize nuclei from frozen skeletal muscle. Effectively, other papers performing snRNASeq from skeletal muscle always use fresh tissue (Nat Commun 2020 Dec 11;11(1):6375. doi: 10.1038/s41467-020-20064-9.; Nat Commun 2020 Oct 9;11(1):5102. doi: 10.1038/s41467-020-18789-8.; Nat Commun 2020 Dec 11;11(1):6374. doi: 10.1038/s41467-020-20063-w.). Otherwise, other papers show nuclei isolation for the same purpose also from frozen tissue of different origin as tumors, kidney, liver, and brain (OMICS 2020 Apr;24(4):180-194. doi: 10.1089/omi.2019.0215. Epub 2020 Mar 16.; J Vis Exp 2021 Sep 20;(175). doi: 10.3791/62901.; J Vis Exp 020 Aug 25;(162). doi: 10.3791/61542.; Nat Med. 2020 May;26(5):792-802. doi: 10.1038/s41591-020-0844-1. Epub 2020 May 11.). Please discuss similarities and differences between the presented protocols and other protocols available in literature. Please specify the ad hoc adaptations that were undertaken to work specifically with human skeletal muscle tissue.

Many cell types have been hypothesized to be involved in muscle disease, such as satellite cells in Duchenne muscular dystrophy (DMD) (Chang et al., 2016), fibroadipogenic progenitors in limb girdle muscular dystrophy (Hogarth et al., 2019; Uezumi et al., 2011); please note that FAPs are strongly involved also in DMD and other myopathies. Please cite proper references

Reviewer #2 (Comments to the Authors (Required)):

In this protocol authors have described an approach they have tested for isolation of nuclei from muscle tissue for the purpose of single nuclear sequencing. It is claimed to be ideal to study banked cryopreserved human muscle samples, including diseased muscles. While this is not the first study of this nature authors claim that this approach is reliable and cost-effective for snRNAseq that uses human skeletal muscle frozen for long periods. The study does not provide any direct comparisons of these features or a direct head-to-head comparison to substantiate these claims, which diminishes the overall validity/impact of the claim.

- The method here is based on tweaks (e.g., beads instead of Dounce to homogenize tissue) over the existing approaches for snRNA preps that have been published, ultimately diminishing the claimed novelty. Even comment such as "too much debris" by Dounce homogenization (versus use of beads) has not been substantiated by any comparison presented in this work.
- Regarding the rapidity, commercial approach (e.g., <https://s2genomics.com/singulator-100/>) for nuclei isolation from muscle (and other tissue) can be done in minutes and has been used for snRNAseq of muscle (medRxiv 2021.01.22.21250336).
- As the vendor (10x) recommends sorting fluorescently labeled nuclei, claimed novelty of Fluorescence assisted nuclei sorting (FANS) is not justified.
- Despite the claimed use of samples frozen for different periods, the authors provide little details regarding dos and don'ts for sample freezing and storage, which is a critical requirement for the success of such an approach.
- Table 4 is disingenuous as it compared scRNAseq approaches with the snRNAseq approach, which are two are entirely different techniques in their scope and strength. Thus, claimed differences between these approaches is moot.
- The stated similarity of QC metrics of different preparative method published compared to their approach identifies that different preparative method give comparable QC. Instead, authors use this to claim their method is superior. Even the number of individual sequences from 6 samples is not significantly greater than the norm.

Overall, I find this is another method of isolating nuclei providing similar quality and quantity as others and belongs more as a methods section of a study instead of a standalone advancement of a protocol.

Reviewer #3 (Comments to the Authors (Required)):

The manuscript by Soule et al describes a protocol for extracting frozen human skeletal muscle nuclei for snRNAseq experiments. The authors show that 60mg of frozen tissue yields a sufficient number of good quality nuclei to allow results with 10X technology.

The reported results are of obvious interest, and the manuscript should be accepted after revisions.

-Little mistake in the abstract, I guess the authors wanted to write "resistant to scRNAseq" instead of snRNAseq.

-I do not understand why the authors presented only results with mixed together all nuclei from all biopsies: impossible to identify healthy and non-healthy samples, mild and major myopathic samples.

I would be curious to see the difference between healthy muscle and the different myopathies. The authors should state which myopathy and show the histological sections, at least an H&E section, which would make clear, whether there are tissular changes that are reflected by the snRNAseq. This would substantiate the paper and validate their approach.

-60 mg tissue for 7000 nuclei is the yield of the experiments, the authors should compare, when possible with existing protocols.

-"To isolate nuclei directly from muscle, these reports used fiber dissection and dounce homogenization" (Dos Santos et al.

2020). "Lyse with mechanical homogenizer at 3000 rpm for 2x5 seconds with a 5 second break in between cycles on ice". What does "mechanical homogenizer" mean? The ceramic beads used to lyse the muscle are not well described or referenced. This must be improved for the reproducibility of the protocol that will be used in many labs.

-I didn't quite understand what is remarkably new about this method compared to published methods? The authors should support the potential improvements in core yield provided by their protocol. In addition, they cite the manuscript by Scripture-Adams et al, who published snRNA-seq data with 3 mg VL human biopsies, but do not mention this manuscript in the first paragraph of the discussion "...has only been reported once before": why?.

- It would have been more interesting to compare the healthy, mild and major profiles in the myonuclei, for example, than the genes associated with the slow and fast program (Figure S5, S6), and to show that relevant genes could be identified in relation to the myopathies studied.

Reviewer #1 (Comments to the Authors):

In the submitted manuscript, Soule and colleagues describe and contextualize a protocol for fast and reliable isolation of nuclei from whole froze human skeletal muscle tissue and performed snRNASeq for sample comparison and quality control.

The protocol steps are well described although some improvements could be done to ameliorate clearness and usage from non-expert users.

INTRODUCTION

Skeletal muscle is a complex tissue composed of around 40% of an adult human's mass: please insert proper citations

Thank you for noticing this. We have inserted an appropriate citation.

It contains many cell types, including satellite cells, myotubes, myogenic precursors, fibroadipogenic progenitors (FAPs), fibroblasts, endothelial cells (ECs), pericytes, adipocytes, immune cells, and smooth muscle.: please insert proper citations

We thank the reviewer for noticing this and have inserted an appropriate citation.

MATERIALS AND METHODS

TABLE 1: please indicate also Date of collection/Freezing of the samples

Thank you very much for requesting this clarification. In our jurisdiction (Alberta, Canada), dates of medical procedures are considered to be identifying information and would require specific consent to disclose. However, we are permitted to disclose the year that the sample was collected and have accordingly modified Table 1.

Lo-Bind 1.5 mL Eppendorf.: this material was not present in the material table

Thank you for noticing this oversight. We have added this to the materials table.

NUCLEI ISOLATION

STEP 10: it is not clear if 200 microliters of Nuclei Wash and Resuspension buffer should be added to the pellet or to the supernatant from step 9. Moreover, since in STEP 8 it has been written that the pellet should contain debris, it is not clear if it would also contain nuclei. Please be more accurate.

Thank you for indicating the lack of clarity. We have elaborated on this step.

STEP 13: which kind of hemocytometer is more suitable for this kind of biological material? Burkner, Thoma, Neubauer? Please also specify some tips to have an accurate counting. How many squares? How to perform mean between squares? How to calculate concentration and dilution?

Thank you for this excellent comment and have added the Neubauer hemocytometer to materials. We also added a comment indicating that the nuclei suspension was undiluted. We also briefly described our counting protocol: "When counting, we average the four outside corner squares of the hemocytometer, including nuclei touching the left and top of each square. However, we recognize that other labs may have their own process for this."

FANS SORTING

STEP 14: Cells will be sitting on ice for extended periods of time without proper protection from RNA degradation - this sentence seems out of context. Which cell the sentence is referring to? At this step of the protocol there should be only nuclei in the suspension. Please rephrase the sentence

Thank you for bringing this to our attention. We have rephrased the sentence.

STEP 14: ***RNase Inhibitor concentration assumes 2 mL final volume of sorted nuclei. Perform preliminary assessments to determine a suitable RNase Inhibitor concentration for sorting. - also this sentence is difficult to understand. please rephrase being more accurate in describing this step

Thank you for this excellent point. We have reworked this portion of the manuscript.

STEP 15: please add a figure about gating strategy

We thank you for this suggestion. Supplementary figure S1 has been added to visualize our gating strategy.

LIBRARY ANALYSIS AND QUALITY CONTROL

In this paragraph you often refer to "cells" rather than to "nuclei". If this protocol is for snRNASeq, it would be better to refer to nuclei. Please also check the entire manuscript. Thank you for this excellent point. We have looked over the document and changed "cells" to "nuclei" where appropriate. This included changing the names of the QC metrics to include nucleus/nuclei instead of cell/cells.

CONSISTENT QUALITY CONTROL ON A VARIETY OF SAMPLES

The average fraction of reads in cells was high (83%), suggesting low ambient RNA in the sample.: please cite and/or refer to other available protocols describing quality parameters in single cells/nuclei sequencing

Thank you for this point. We have added a reference describing several aspects of quality control. "Given this, our average fraction of reads in nuclei was high (83%), suggesting low ambient RNA in the sample (Hong et al., 2022)."

DISCUSSION

The manuscript describes a protocol to isolate and characterize nuclei from frozen skeletal muscle. Effectively, other papers performing snRNASeq from skeletal muscle always use fresh tissue (Nat Commun 2020 Dec 11;11(1):6375. doi: 10.1038/s41467-020-20064-9.; Nat Commun 2020 Oct 9;11(1):5102. doi: 10.1038/s41467-020-18789-8.; Nat Commun 2020 Dec 11;11(1):6374. doi: 10.1038/s41467-020-20063-w.). Otherwise, other papers show nuclei isolation for the same purpose also from frozen tissue of different origin as tumors, kidney, liver, and brain (OMICS 2020 Apr;24(4):180-194. doi: 10.1089/omi.2019.0215. Epub 2020 Mar 16.; J Vis Exp 2021 Sep 20;(175). doi: 10.3791/62901.; J Vis Exp 2020 Aug 25;(162). doi: 10.3791/61542.; Nat Med. 2020 May;26(5):792-802. doi: 10.1038/s41591-020-0844-1. Epub 2020 May 11.). Please discuss similarities and differences between the presented protocols and other protocols available in literature. Please specify the ad hoc adaptations that were undertaken to work specifically with human skeletal muscle tissue.

We have added a brief discussion surrounding the protocols associated with isolating nuclei from different tissues. “Each tissue requires specific protocol adaptations to maximize the yield and quality of nuclei (Eraslan et al., 2022; Slyper et al., 2020). Common elements that vary between protocols include buffers, instruments of dissociation, and methods of purification. This is emphasized by the distinct protocols developed to isolate nuclei from liver (Cavalli et al., 2020), kidney (Leiz et al., 2021), and tumours (Narayanan et al., 2020; Slyper et al., 2020). We have developed a protocol that works for the dense, fibrous nature of skeletal muscle.”

Many cell types have been hypothesized to be involved in muscle disease, such as satellite cells in Duchenne muscular dystrophy (DMD) (Chang et al., 2016), fibroadipogenic progenitors in limb girdle muscular dystrophy (Hogarth et al., 2019; Uezumi et al., 2011); please note that FAPs are strongly involved also in DMD and other myopathies. Please cite proper references
Thank you for raising this excellent point. We have added references pertaining to the role of FAPs in DMD and other myopathies.

Reviewer #2 (Comments to the Authors):

In this protocol authors have described an approach they have tested for isolation of nuclei from muscle tissue for the purpose of single nuclear sequencing. It is claimed to be ideal to study banked cryopreserved human muscle samples, including diseased muscles. While this is not the first study of this nature authors claim that this approach is reliable and cost-effective for snRNAseq that uses human skeletal muscle frozen for long periods. The study does not provide any direct comparisons of these features or a direct head-to-head comparison to substantiate these claims, which diminishes the overall validity/impact of the claim.

Thank you very much for pointing this out. We completely agree that it would have been ideal to compare two or more methods head-to-head. We were unable to do this because human skeletal muscle is a non-renewable and limited resource. It was not in the scope of our project to do a comparison of methods, rather, our goal was to develop a method that would be reproducible and to carefully document quality control metrics. However, we acknowledge this very important point and have added to the discussion to raise this issue: “In the course of future study, it would be informative to perform a direct comparison of different methods, to more clearly outline the advantages of each protocol without inter-operator variability.”

- The method here is based on tweaks (e.g., beads instead of Dounce to homogenize tissue) over the existing approaches for snRNA preps that have been published, ultimately diminishing the claimed novelty. Even comment such as "too much debris" by Dounce homogenization (versus use of beads) has not been substantiated by any comparison presented in this work. We thank the reviewer for raising this point. We have added figure S8 to compare the debris yielded from the Dounce and Bertin homogenizers in our hands. Additionally, in Table 4 we have added a comment regarding the inherent inter-operative variability of dounce homogenization compared with consistent timing and shaking speed of the Bertin homogenizer in our method.

- Regarding the rapidity, commercial approach (e.g., <https://s2genomics.com/singulator-100/>) for nuclei isolation from muscle (and other tissue) can be done in minutes and has been used for snRNAseq of muscle (medRxiv 2021.01.22.21250336).

We thank the reviewer for this comment. Our intent was to report a protocol that can be used by labs without access to expensive machinery. We acknowledge that there will be some variability between those using this protocol, however, we still believe that our method will be reliable, fast, and useful in the analysis of biobanked samples. We noted that the provided medrxiv publication has just been published in *Aging* and have provided a citation to this important work and referred the reader to the proprietary tool in the introduction: "One recently published work used fresh muscle samples with a proprietary instrument specifically designed for sc/snRNAseq (S2 Genomics, Livermore, CA, USA)."

- As the vendor (10x) recommends sorting fluorescently labeled nuclei, claimed novelty of Fluorescence assisted nuclei sorting (FANS) is not justified.

Thank you for this point. To this end, we have added an acknowledgement that 10X recommends FANS: "As recommended by 10X Genomics, fluorescence assisted nuclei sorting (FANS) was used to purify nuclei because it minimally impacts RNA expression and integrity compared to enzymatic dissociation. The use of enzymes like collagenase and dispase are widely used to dissociate muscle tissue for single cell analysis, but have been reported to affect transcription in single cells (Eraslan et al., 2022; Orchard et al., 2021; Van Den Brink et al., 2017)."

- Despite the claimed use of samples frozen for different periods, the authors provide little details regarding dos and don'ts for sample freezing and storage, which is a critical requirement for the success of such an approach.

Thank you for pointing out that further detail would have been beneficial. We have included the following in the discussion: "We recommend that muscle samples used for this protocol should be flash-frozen after devitalisation and stored continuously in a sealed container at -80oC thereafter. Storage at temperatures above -80oC or in an unsealed container may affect sample integrity and should be avoided. Freeze-thaw cycles are also expected to negatively impact sample integrity. However, we did not formally test these recommendations in this work and it is possible that samples with suboptimal storage could yield useable results in the course of future study."

- Table 4 is disingenuous as it compared scRNAseq approaches with the snRNAseq approach, which are two entirely different techniques in their scope and strength. Thus, claimed differences between these approaches is moot.

We thank you for your analysis of this section. We agree that snRNAseq and scRNAseq differ in some aspects. However, Table 4 highlights specific aspects of our protocol that are analogous to scRNAseq, therefore, we believe that these points are relevant. In the introduction, we discuss our justification for choosing to isolate nuclei instead of single cells, as well as the differences between single cell and single nucleus derived data. However to bring attention to the reviewer's excellent point we have added the following when introducing Table 4: "with the caveat that methods and results from single cell and single nucleus isolation will not always be directly comparable."

- The stated similarity of QC metrics of different preparative method published compared to their approach identifies that different preparative method give comparable QC. Instead, authors use this to claim their method is superior. Even the number of individual sequences from 6 samples is not significantly greater than the norm. We thank you for this observation. It was not our intention to claim superiority as it is difficult to compare QC metrics when they are rarely reported in full. We were mindful of this and tried to reflect that in the manuscript: "Although quality control metrics are rarely reported in full, our values were similar to previous studies (Barruet et al., 2020; Dos Santos et al., 2020; Kim et al., 2020; Petrany et al., 2020; Scripture-Adams et al., 2022). In summary, quality control metrics meet 10X Genomics' recommendations and are in line with values from similar studies."

We also changed wording in the introduction to "excellent yield *comparable* to other methods". We can appreciate that the original wording ("compared") may have implied superiority, which was not our intent. We are grateful to the reviewer for raising this important point.

Overall, I find this is another method of isolating nuclei providing similar quality and quantity as others and belongs more as a methods section of a study instead of a standalone advancement of a protocol.

We thank the reviewer for their expertise and detailed review of our work. We submitted the following work as a standalone protocol to provide the high level of detail needed for a non-expert user to carry out this experiment. We believe that such protocol documents are valuable to the research community for highly topical methods such as snRNAseq, and that it will improve uptake and adoption of novel methods to research programs.

Reviewer #3 (Comments to the Authors):

The manuscript by Soule et al describes a protocol for extracting frozen human skeletal muscle nuclei for snRNAseq experiments. The authors show that 60mg of frozen tissue yields a sufficient number of good quality nuclei to allow results with 10X technology.

The reported results are of obvious interest, and the manuscript should be accepted after revisions.

-Little mistake in the abstract, I guess the authors wanted to write " resistant to scRNAseq " instead of snRNAseq.

Thank you for pointing out this oversight. The abstract wording has been changed.

-I do not understand why the authors presented only results with mixed together all nuclei from all biopsies: impossible to identify healthy and non-healthy samples, mild and major myopathic samples.

I would be curious to see the difference between healthy muscle and the different myopathies. The authors should state which myopathy and show the histological sections, at least an H&E section, which would make clear, whether there are tissular changes that are reflected by the snRNAseq. This would substantiate the paper and validate their approach.

Thank you very much for noticing this and for requesting clarification. The goal of this work was to present a protocol that would be effective for snRNAseq of frozen human muscle tissues, for both diseased and control samples. The comparison of control and myopathy phenotypes (both clinical and histopathologic) is certainly the goal of future and ongoing study in our lab. For purposes of this protocol publication, we were concerned that the small sample size would make any comparisons potentially misleading and could be contradicted as we carried out future studies with larger sample sizes. However, we very much appreciate the thoughtful recommendation.

-60 mg tissue for 7000 nuclei is the yield of the experiments, the authors should compare, when possible with existing protocols.

We thank the reviewer for their suggestion. There are some challenges that prevent us from comparing this metric with other papers. The number of nuclei that are recovered for data processing depends on the number of nuclei that are loaded into the 10X machine. The total number of nuclei recovered from a given tissue is rarely reported. Additionally, some groups may want to load more or less nuclei depending on their experimental aims. For example, our method yields significantly more than 7000 nuclei from the 60 mg of tissue, however, we only load 15000 nuclei to target a recovery of 10000 nuclei. For this reason we did not include comparisons with other protocols but perhaps it can be considered as future protocol papers are published.

- "To isolate nuclei directly from muscle, these reports used fiber dissection and dounce homogenization" (Dos Santos et al. 2020). "Lyse with mechanical homogenizer at 3000 rpm for 2x5 seconds with a 5 second break in between cycles on ice". What does "mechanical homogenizer" mean? The ceramic beads used to lyse the muscle are not well described or referenced. This must be improved for the reproducibility of the protocol that will be used in many labs.

We thank you for these excellent comments. We have changed wording for "mechanical homogenizer" to "Secure bead tube in mechanical homogenizer and shake at 3000 rpm for 2x5 seconds with a 5 second break in between cycles on ice." Also, the ceramic beads are

referenced under “consumables” in table 2. The beads arrive pre-packaged in a tube, ready for use.

-I didn't quite understand what is remarkably new about this method compared to published methods? The authors should support the potential improvements in core yield provided by their protocol.

Thank you very much for requesting clarification. As stated in our paper, there are certain specific advantages of this method (works on frozen tissue, reproducibility, ease and speed of processing, and was tested in an archival sample, all with comparable quality control metrics to other published methods). We do not necessarily believe the method is superior to other described protocols (see also our responses to Reviewer 2). However we believe the strength of this work lies in the detailed description of the protocol and its particular advantages which we also summarise in Table 4. We believe an important value of this work will be in supporting the development of snRNAseq methods in other labs who are looking to add this to their projects.

In addition, they cite the manuscript by Scripture-Adams et al, who published snRNA-seq data with 3 mg VL human biopsies, but do not mention this manuscript in the first paragraph of the discussion "...has only been reported once before": why?.

We thank you for pointing out this oversight. We have added references for Eraslan et al., 2022, and Scripture-Adams et al., 2022 in this sentence.

- It would have been more interesting to compare the healthy, mild and major profiles in the myonuclei, for example, than the genes associated with the slow and fast program (Figure S5, S6), and to show that relevant genes could be identified in relation to the myopathies studied. Thank you for this excellent recommendation. We agree completely with this reviewer that a detailed analysis of controls and samples with mild or major pathologic changes would be of value. This is part of current and future ongoing work in our lab and is outside the scope of the current protocol paper. We considered including such analysis in this paper however we were concerned results may be incorrect or misleading based on limited sample size. We are planning to include the analyses that this reviewer has astutely recommended in future publications.

January 20, 2023

Re: Life Science Alliance manuscript #LSA-2022-01806-TR

Dr. Gerald Pfeffer
University of Calgary
Hotchkiss Brain Institute, Department of Clinical Neurosciences
HMRB 155, 3330 Hospital Dr NW
Calgary, Alberta T2N 4N1
Canada

Dear Dr. Pfeffer,

Thank you for submitting your revised manuscript entitled "A Protocol for Single Nucleus RNAseq from Frozen Skeletal Muscle" to Life Science Alliance. The manuscript has been seen by the original reviewers whose comments are appended below. While Reviewers 1 & 3 continue to be overall positive about the work in terms of its suitability for Life Science Alliance, some important issues remain for Reviewer 2 that need to be addressed before this manuscript can be considered for publication at LSA.

Our general policy is that papers are considered through only one revision cycle; however, given that the suggested changes are relatively minor, we are open to one additional short round of revision. Please note that I will expect to make a final decision without additional reviewer input upon resubmission.

Please submit the final revision within one month, along with a letter that includes a point by point response to the remaining reviewer comments.

To upload the revised version of your manuscript, please log in to your account: <https://lsa.msubmit.net/cgi-bin/main.plex>
You will be guided to complete the submission of your revised manuscript and to fill in all necessary information.

B. MANUSCRIPT ORGANIZATION AND FORMATTING:

Sincerely,

Reviewer #1 (Comments to the Authors (Required)):

In the presented manuscript, the authors report about a method to obtain and analyze single-nuclei preparation from frozen skeletal muscle specimens obtained from human biopsies.

Although the novelty of the method is not fully justified, the manuscript is well-presented and the authors fulfilled the required revision.

Given these premises, this Reviewer does not see any reason of rejection.

Reviewer #2 (Comments to the Authors (Required)):

The revised manuscript is improved over the previous submission. It has toned down claims related to superiority of this approach over the published work. However, to make this a publishable standalone protocol the approach needs to be made superior, which is still lacking. Authors acknowledge that "it would be informative to perform a direct comparison of different methods", but instead of such a comparison and improvements to address deficits, authors state future studies are needed for this.

The manuscript states this is a "fast, and reliable method to isolate over 7000 nuclei on average" is misleading as the diseased (M1-M4) tissues provide an average of 5000 nuclei (ranging from 1000 - 8500 nuclei), while the two healthy muscles used provide twice the nuclear yield ~11,000 nuclei driving up the stated average. This 10-fold difference in yield (1000 - 13,000 nuclei) and selective bias against diseased muscles, diminishes the claimed value for this approach.

Contrary to the author claim that "our values were similar to previous studies (Barruet et al., 2020; Dos Santos et al., 2020; Kim et al., 2020; Petrany et al., 2020; Scripture-Adams et al., 2022).", reported nuclear yields appear to be higher 6000 - 60,000 (Scripture-Adams et al), 200,000 (Dos Santos et al). Even the claim regarding value of FANS, authors acknowledge that this is a vendor recommendation and they are able to meet this requirement, not provide an advancement.

Table 4 is used to claim superiority of snRNAseq over scRNAseq methodology, but authors acknowledge this cannot be done as "methods and results from single cell and single nucleus isolation will not always be directly comparable." The table also fails to list the many deficits of the snRNA approach regarding depth of RNA capture, failure to capture steady state cytosolic mRNAs (in favor of active nuclear transcription), excess unprocessed RNAs etc. which would impact of study design or use of this protocol.

To address the need to specify sample storage condition for improved snRNAseq authors have now stated their recommendation, but then clarify "we did not formally test these recommendations in this work", which makes their recommendation a speculation, not a technological advance from this work.

In summary, I find that similar to the original manuscript, the revised manuscript does not still offer sufficient advances for a new protocol.

Reviewer #3 (Comments to the Authors (Required)):

The manuscript has been improved and may be published.

Reviewer #1 (Comments to the Authors (Required)):

In the presented manuscript, the authors report about a method to obtain and analyze single-nuclei preparation from frozen skeletal muscle specimens obtained from human biopsies.

Although the novelty of the method is not fully justified, the manuscript is well-presented and the authors fulfilled the required revision.

Given these premises, this Reviewer does not see any reason of rejection.

Reviewer #2 (Comments to the Authors (Required)):

The revised manuscript is improved over the previous submission. It has toned down claims related to superiority of this approach over the published work. However, to make this a publishable standalone protocol the approach needs to be made superior, which is still lacking. Authors acknowledge that "it would be informative to perform a direct comparison of different methods", but instead of such a comparison and improvements to address deficits, authors state future studies are needed for this.

We appreciate the reviewer's comment and are very grateful for their critical appraisal of our work. We have presented this protocol as an alternative approach for preparation of frozen human skeletal muscle for snRNAseq. We believe there are some important advantages to the method (including cost of the approach and the reproducibility of the method), which we have outlined, and have provided detailed quality control data.

It was not our goal to produce a method that was necessarily superior, but rather to optimize a method that would be useful for clinical researchers who have human muscle samples in storage and are considering snRNAseq studies. Some researchers may consider our method superior for their needs and some may not. We have already had enquiries about the method from other researchers based on our preprint in Biorxiv. We believe there is strong value in the publication of alternative protocols, especially with highly topical methods such as snRNAseq.

Future studies would be required to directly compare methods. In our case, such a comparison would not be possible because of the non-renewable nature of the muscle tissues we have access to. However we are very appreciative of the reviewer's important comment and this is why we included the recommendation for additional studies in the manuscript.

The manuscript states this is a "fast, and reliable method to isolate over 7000

nuclei on average" is misleading as the diseased (M1-M4) tissues provide an average of 5000 nuclei (ranging from 1000 - 8500 nuclei), while the two healthy muscles used provide twice the nuclear yield ~11,000 nuclei driving up the stated average. This 10-fold difference in yield (1000 - 13,000 nuclei) and selective bias against diseased muscles, diminishes the claimed value for this approach.

We thank the reviewer for raising this point. In the manuscript we acknowledge that the muscles with pathological changes recover less nuclei (1161, 3944, 6997, 8531). Both of our controls recovered our target of around 10,000 nuclei (9697, 13030). It is probable that these differences are biological (i.e: due to the presence of increased fibroadipose tissues in diseased muscles). Despite this, our QC data demonstrate that the recovered nuclei from the disease conditions display very similar metrics to that of the controls. Therefore, we believe that although some disease conditions may reduce the overall nuclear yield, recovered nuclei still hold the potential to yield valuable biological insights.

To address this very important and astute point, we have modified the text accordingly in Results:

Although we targeted 10,000 nuclei, the number of recovered nuclei after sequencing varied. Our two controls recovered around 10,000 nuclei, with the disease samples yielding less (respectively, 1161, 3944, 6997, 8531). This is likely due to the increased fibroadipose tissues that are contained within diseased muscle. Despite this, quality control metrics were similar between diseased muscle and controls. Furthermore~~Although quality control metrics are rarely reported in full~~, our values were similar to previous studies (Barruet et al., 2020; Dos Santos et al., 2020; Kim et al., 2020; Petrany et al., 2020; Scripture-Adams et al., 2022), although quality control metrics were not always reported in full. In summary, quality control metrics meet 10X Genomics' recommendations and are in line with values from similar studies.

Contrary to the author claim that "our values were similar to previous studies (Barruet et al., 2020; Dos Santos et al., 2020; Kim et al., 2020; Petrany et al., 2020; Scripture-Adams et al., 2022).", reported nuclear yields appear to be higher 6000 - 60,000 (Scripture-Adams et al), 200,000 (Dos Santos et al). Even the claim regarding value of FANS, authors acknowledge that this is a vendor recommendation and they are able to meet this requirement, not provide an advancement.

We thank the reviewer for addressing this topic. Although the referenced papers do isolate many nuclei, these numbers represent the total nuclei from each sample. Only a fraction of these are loaded onto the 10X Chromium Controller, and this number varies depending on the goal of each study. For this reason, we only

reported the number of nuclei loaded onto the 10X Chromium controller and the amount recovered after 10X processing and QC.

To illustrate this point, Scripture-Adams isolated 6,000-60,000 nuclei per sample, pooled their samples, and only loaded 10,000-20,000 nuclei onto the 10X Genomics Chromium Controller. 35% of those resulted in viable libraries. The result was 7549 nuclei from 5 human muscle samples, an average of just over 1500 nuclei per sample. Dos Santos and colleagues isolated 200,000 nuclei from mouse muscle and loaded 4000 nuclei onto the 10X Genomics Chromium Controller. To our knowledge, Barruet and colleagues did not report their total number of isolated satellite cells. Instead, they detail loading 18,000 human satellite cells and recovering 8,500 cells per sample. Perez and colleagues started with 20-50 mg of human tissue, but do not report number of nuclei isolated or loaded for GEM generation. They recover around 8,400 nuclei per sample. In our study, we loaded around 16,000 nuclei, and recovered just over 7000 nuclei on average. In summary, we demonstrate that we recover sufficient nuclei to perform snRNAseq and our values are in line with previous literature reports.

Additionally, to improve clarity for other researchers who may try this protocol, we have added to the results section: "We isolated between 12,000 and 27,000 nuclei per sample with an average of 19,500. Only two samples yielded less than 16,000 nuclei, in which case we loaded the full suspension volume onto the 10X Chromium Controller."

Table 4 is used to claim superiority of snRNAseq over scRNAseq methodology, but authors acknowledge this cannot be done as "methods and results from single cell and single nucleus isolation will not always be directly comparable." The table also fails to list the many deficits of the snRNA approach regarding depth of RNA capture, failure to capture steady state cytosolic mRNAs (in favor of active nuclear transcription), excess unprocessed RNAs etc. which would impact of study design or use of this protocol.

We thank the reviewer for this opportunity to clarify. Our intention in Table 4 was to justify our methodological choices given the constraints of previously frozen and banked muscle tissue. Table 4 illustrates the advantage of employing snRNAseq instead of scRNAseq, why we chose FANS, and why we chose the bead tube lysis. However, we agree that differences between transcripts present in cells and nuclei should be addressed. To this end, we have added two references discussing how snRNAseq contains a higher proportion of introns, but no significant differences in global transcript profiles: "Single nuclear RNA sequencing (snRNAseq) has been demonstrated by multiple groups to yield comparable data to scRNAseq (Slyper et al., 2020; Zeng et al., 2016). snRNAseq recovers the same cell types present in a tissue, but in different proportions (Slyper et al., 2020). Notably, there are no significant differences between gene expression profiles of nuclei and whole cells (Barthelson,

Lambert, Vanier, Lynch, & Galbraith, 2007; Grindberg et al., 2013), with the exception of a few specific processes like cell cycle, mitosis, and transcription (Barthelson et al., 2007). Also, nuclei are enriched for non-coding RNAs (Zeng et al., 2016) and contain a higher proportion of introns (Grindberg et al., 2013)."

To address the need to specify sample storage condition for improved snRNAseq authors have now stated their recommendation, but then clarify "we did not formally test these recommendations in this work", which makes their recommendation a speculation, not a technological advance from this work. In summary, I find that similar to the original manuscript, the revised manuscript does not still offer sufficient advances for a new protocol.

We thank the reviewer for noticing this limitation. Although we did not formally test the recommendation, the recommendation is made based on our procedures for this study. We believe it is still useful for readers to know what was done, and what we recommend based on how our samples were handled. In fact, we have already had an enquiry along these lines to our Biorxiv preprint, which had not included this information. So, we are very thankful that the reviewer recommended that we include this revision in the manuscript and we believe it will be very useful to readers.

We are not able to test other sample handling approaches (e.g: freeze-thaw cycles) because of the non-renewable and limited quantity of our human muscle samples. If other researchers have the ability and interest to test this, that would be an important contribution and hopefully they will publish those results or be in contact with us.

We are not aware of any prior research that investigated the effect of sample handling or freeze thaw cycles prior to snRNAseq. However, to partially address this reviewer's important question we have added two citations to support our recommendation, one showing the impact of freeze thaw cycles on RNAseq (Kellman et al), and another suggesting suboptimal sample handling may not affect RNA quality (de Oliveira et al).

Reviewer #3 (Comments to the Authors (Required)):

The manuscript has been improved and may be published.

February 8, 2023

RE: Life Science Alliance Manuscript #LSA-2022-01806-TRR

Dr. Gerald Pfeffer
University of Calgary
Hotchkiss Brain Institute, Department of Clinical Neurosciences
HMRB 155, 3330 Hospital Dr NW
Calgary, Alberta T2N 4N1
Canada

Dear Dr. Pfeffer,

Thank you for submitting your revised manuscript entitled "A Protocol for Single Nucleus RNAseq from Frozen Skeletal Muscle". We would be happy to publish your paper in Life Science Alliance pending final revisions necessary to meet our formatting guidelines.

- please upload your main figures as single files; these will be displayed in-line in the HTML version of your paper, so please provide them as single page files (Figure 1 spans 2 pages); we do not have a limit on the number of main figures and these can be split if necessary for space
- please consult our manuscript preparation guidelines <https://www.life-science-alliance.org/manuscript-prep> and make sure your manuscript sections are in the correct order
- please add an Author Contributions section to your main manuscript text
- please add your main, supplementary figure, and table legends to the main manuscript text after the references section
- please add callouts for Figures S7 and S8A-B to your main manuscript text

A. FINAL FILES:

B. MANUSCRIPT ORGANIZATION AND FORMATTING:

Sincerely,

February 27, 2023

RE: Life Science Alliance Manuscript #LSA-2022-01806-TRRR

Dr. Gerald Pfeffer
University of Calgary
Hotchkiss Brain Institute, Department of Clinical Neurosciences
HMRB 155, 3330 Hospital Dr NW
Calgary, Alberta T2N 4N1
Canada

Dear Dr. Pfeffer,

Thank you for submitting your Methods entitled "A Protocol for Single Nucleus RNAseq from Frozen Skeletal Muscle". It is a pleasure to let you know that your manuscript is now accepted for publication in Life Science Alliance. Congratulations on this interesting work.

DISTRIBUTION OF MATERIALS:

Again, congratulations on a very nice paper. I hope you found the review process to be constructive and are pleased with how the manuscript was handled editorially. We look forward to future exciting submissions from your lab.

Sincerely,
